# Integrating In Vitro and In Silico Approaches to Assess *Monotheca buxifolia* Plant Extract against *Rhipicephalus* (*Boophilus*) *microplus* and *Sarcoptes scabiei*

**DOI:** 10.3390/molecules28196930

**Published:** 2023-10-04

**Authors:** Afshan Khan, Salman Taj, Nosheen Malak, Ayman A. Swelum, Adil Khan, Nasreen Nasreen, Sadaf Niaz, Wen-Feng Wu

**Affiliations:** 1Department of Zoology, Abdul Wali Khan University Mardan, Mardan 23200, Pakistan; afshankhan230@gmail.com (A.K.); salmantaj308@gmail.com (S.T.); nosheenmalak496@gmail.com (N.M.); nasreen@awkum.edu.pk (N.N.); sadaf@awkum.edu.pk (S.N.); 2Department of Animal Production, College of Food and Agriculture Sciences, King Saud University, Riyadh 11451, Saudi Arabia; aswelum@ksu.edu.sa; 3Department of Biology, Mount Allison University, Sackville, NB E4L 1G7, Canada; 4Department of Botany & Zoology, Bacha Khan University Charsadda, Charsadda 24420, Pakistan; 5Department of Radiology, Ditmanson Medical Foundation Chia-Yi Christian Hospital, Chiayi 60002, Taiwan

**Keywords:** *Sarcoptes scabiei* aspartic protease (SsAP), *Rhipicephalus microplus* aspartic protease (RmAP), docking, ectoparasites, *Monotheca buxifolia* extract, acaricide resistance, molecular dynamics simulations

## Abstract

Tick and mite infestations pose significant challenges to animal health, agriculture, and public health worldwide. The search for effective and environmentally friendly acaricidal agents has led researchers to explore natural alternatives. In this study, we investigated the acaricidal potential of the *Monotheca buxifolia* plant extract against *Rhipicephalus microplus* ticks and *Sarcoptes scabiei* mites. Additionally, we employed a computational approach to identify phytochemicals from the extract that could serve as drug candidates against these ectoparasites. The contact bioassay results demonstrated that the *M. buxifolia* plant extract exhibited significant efficacy against *R. microplus* and *S. scabiei*, with higher concentrations outperforming the positive control acaricide permethrin in terms of mite mortality. Time exposure to the extract also showed a positive correlation with better lethal concentration (LC_50_ and LC_90_) values. Similarly, the adult immersion test revealed a notable inhibition of tick oviposition via the plant extract, especially at higher concentrations. The two-protein primary structure, secondary structure and stability were predicted using the Expasy’s ProtParam server, SOPMA and SUSUI server, respectively. Using Homology modeling, the 3D structure of the protein was obtained and validated through the ERRAT server, and active sites were determined through the CASTp server. The docking analysis revealed that Alpha-Amyrenyl acetate and alpha-Tocopherol exhibited the highest docking scores for *S. scabiei* and *R. microplus* aspartic protease proteins, respectively. These phytochemicals demonstrated strong binding interactions, suggesting their potential as acaricidal drug candidates. In conclusion, the *M. buxifolia* plant extract displayed significant acaricidal activity against *R. microplus* and *S. scabiei.* Moreover, the computational approach identified promising phytochemicals that could serve as potential drug candidates for controlling these ectoparasites.

## 1. Introduction

Sarcoptic mange, also known as scabies in human patients, is an extremely contagious skin condition caused by *Sarcoptes scabiei*, an astigmatic mite that burrows into the epidermis, actively penetrating the stratum corneum [1]. The adult mites mate, and the females lay eggs in the skin. The hatched larvae create small burrows, referred to as molting pouches, where they molt into nymphs and eventually mature into adults. These parasites have a worldwide distribution, affecting over 150 host species, and they demonstrate remarkable epidemiological adaptability, enabling transmission among diverse hosts [2]. The disease can also manifest as a mild infection in animals, presenting symptoms such as itching papules, erythema, scales, and alopecia. In chronic cases, hyperkeratosis and/or exudative crust formation may be observed [3,4].

Sarcoptic mange is an exceptionally contagious skin disease that spreads through direct skin-to-skin contact, fomites or contact with contaminated environments frequented by heavily affected hosts [5,6]. Nymphs and females have remarkable survival capabilities off the host, lasting up to 21 days, and exhibit higher resistance compared to larvae and males [7,8]. Thus, the implementation of biocides and repellents in the environment becomes paramount. Wildlife, unfortunately, is particularly susceptible to sarcoptic mange, and its outbreaks can lead to heightened morbidity and even fatal consequences [9], especially when naive populations are affected [10]. This disease is rapidly emerging [11] and has been implicated in the decline of wildlife populations [12,13], resulting in reduced reproduction rates and triggering mass mortality events [14,15].

The *Rhipicephalus microplus* tick (Acari: *Ixodidae*) is a profoundly impactful tick species with devastating effects on bovine well-being and productivity. Its widespread presence in various tropical and subtropical regions worldwide, particularly in Pakistan and India [16], poses a severe threat to cattle populations. Infestations by these ticks lead to blood loss, resulting in anemia and substantial economic losses due to reduced growth and production. Moreover, these ticks serve as vectors for various diseases, including *Babesia bigemina* and *Anaplasma marginale*, which cause economically significant and very deadly illnesses, particularly in high-yielding crossbred and exotic cattle herds [17]. Ticks and tick-borne illnesses generate an estimated USD 13.9 to 18.7 billion loss in cattle alone, with an annual deficit of nearly 3 billion pieces of hide and skin [18]. Therefore, it is of the highest significance to create highly efficient ways for managing cattle ticks and guaranteeing optimum health and productivity in dairy animals.

*Monotheca buxifolia*, a member of the *Sapotaceae* family, is a very important indigenous evergreen tree or thorny shrub known for its tasty fruits that grows in a variety of habitats including northwest Pakistan, Afghanistan, and Iran. This excellent plant performs a variety of essential functions, including feed, fuel, tiny wood, and fences. Its importance extends to traditional medicine, where the fruit has been valued for its digestive, hematinic, vermicide, and laxative characteristics, and the aerial portions are used to treat rheumatism and urinary tract infections [19]. The plant extracts have been scientifically proven to have a wide range of beneficial properties, including “hepatoprotective, urease inhibitory, reno-protective, antidepressant, antiproliferative, antibacterial, antioxidant, anxiolytic, antimalarial, analgesic, antipyretic, and anti-inflammatory activities” [19,20,21,22,23,24].

Aspartic protease is an enzymatic protein responsible for catalyzing the hydrolysis of peptide bonds with the aid of a trapped water molecule [25]. These proteases contain a pair of aspartic acid residues at their active site, facilitating various crucial physiological processes in parasitic organisms, including “tissue invasion, migration, digestion, molting, and reproduction”. For ticks, hemoglobin serves as a primary nutrient that is crucial for their development and reproduction. Consequently, the digestion of hemoglobin is a vital metabolic pathway occurring within digestive vesicles absorbed into gut epithelial cells through the action of aspartic proteases [26]. In a *S. scabiei* cDNA library, researchers identified a single aspartic protease, and functional assays revealed its ability to digest human “hemoglobin, serum albumin, fibronectin, and fibrinogen” while not affecting collagen III or laminin. The presence of this aspartic protease (SsAP) opens up possibilities for interfering with its function, which may have implications for mite survival [27].

In this context, the present study aimed to assess the potential of *M. buxifolia* plant extract to inhibit ticks and mites in vitro, as well as to investigate the inhibitory interactions between the plant’s phytochemical and the aspartic protease proteins of ticks and mites using in silico methods. It is worth noting that, prior to our research, there has been limited exploration of *M. buxifolia* plant extract for its acaricidal potential against these ticks and mites. Furthermore, the specific phytochemicals derived from this plant have not been previously utilized in docking experiments aimed at understanding their interactions with the target proteins in ticks and mites. This novelty adds significance to our study, as it contributes to the emerging knowledge about the potential use of *M. buxifolia* in pest control and sheds light on the underlying molecular mechanisms.

## 2. Results

### 2.1. Contact Bioassay

The efficacy of the *M. buxifolia* plant extract against the *S. scabiei* var. *cuniculi* at varying concentrations were tested in different time intervals, as shown in Table 1 and Figure 1 and Figure 2, along with the positive control, permethrin, and the negative control, distilled water. The higher concentrations of 1 and 2 g/mL outperformed the positive control, acaricide permethrin, in terms of the mites’ mortality. The theoretical lethal concentrations responsible for the 50% and 90% mortality of the test mites’ population were calculated according to Finney [28] calculation. The higher time exposure of the mites to the extract resulted in better LC_50_ and LC_90_ values as shown in Table 2 and Figure 3. At 6 h treatment, The LC_50_ value and its corresponding calculated confidence limits were 0.342 (0.202–0.467) g/mL, and the LC_90_ value and its corresponding confidence limits were 1.529 (1.037–3.429) g/mL. Similarly, the lethal time required to kill 50% and 90% (LT_50_ and LT_90_) of the test mites’ populations at an extract concentration of 2 g/mL was 0.931 (0.673–1.192) and 3.356 (2.474–5.479) h, respectively, as shown in Table 3, Figure 4.

### 2.2. Adult Immersion Test (AIT)

In the adult immersion test (AIT), the efficacy of *M. buxifolia* plant extract in inhibiting the oviposition (IO) of ticks was assessed at different concentrations. The results revealed a significant inhibition of egg laying by ticks when exposed to the extract. At a 40 mg/mL concentration, the extract displayed a substantial IO of 35.612%, demonstrating its potential to hinder the reproductive capabilities of ticks. Lower concentrations of 20 mg/mL, 10 mg/mL, 5 mg/mL, and 2.5 mg/mL also exhibited notable IO values of 30.574%, 15.960%, 10.239%, and 1.326%, respectively (Figure 2, Table 4). In comparison, the control group treated with permethrin (3 mg/mL) exhibited the highest IO of 54.755%, reaffirming its effectiveness as a tick repellent. Conversely, the control group treated with distilled water displayed a negative IO of −6.386%, indicating no inhibitory effects on egg laying. These findings highlight the potential of the *M. buxifolia* extract, particularly at higher concentrations, as a promising natural alternative for controlling tick populations by disrupting their reproductive processes.

### 2.3. Larval Packet Test (LPT)

Table 5 provides the medium lethal concentration (LC_50_ and LC_90_) values of *M. buxifolia* plant extract against *R. microplus* in vitro using the AIT. At 24 h, the extract exhibited an LC_50_ value of 48.678 mg/mL, indicating the concentration required to cause 50% mortality, while the LC_90_ value was 85.498 mg/mL, representing the concentration required for 90% mortality. At 48 h, the extract showed a significantly lower LC_50_ value of 3.013 mg/mL and an LC_90_ value of 43.759 mg/mL, indicating its increased potency over time. The statistical analysis confirmed significant differences in effectiveness between the concentrations and time points (*p* < 0.05).

Table 6 and Figure 2 present the lethal time (LT_50_ and LT_90_) values for *M. buxifolia* extract against *R. microplus* in vitro using the AIT. The LT_50_ values ranged from 24.77 h (at 40 mg/mL concentration) to 60.179 h (at 2.5 mg/mL concentration), signifying the time required to achieve 50% mortality. Similarly, the LT_90_ values ranged from 49.446 h (at 40 mg/mL concentration) to 226.023 h (at 2.5 mg/mL concentration), representing the time needed to attain 90% mortality. The statistical analysis did not reveal significant differences between the concentrations (*p* > 0.05).

These results indicate that the *M. buxifolia* plant extract exhibited concentration-dependent effects on the lethal concentrations and lethal time in the AIT against *R. microplus*. The extract demonstrated significant potency in inducing mortality, with lower concentrations and longer exposure times showing greater effectiveness.

### 2.4. Primary Structure Prediction

The primary structure of aspartic protease proteins from *R. microplus* and *S. scabiei* mite was analyzed using the highly reliable Expasy ProtParam server. The *S. scabiei* aspartic protease protein was found to be 419 amino acids long with a molecular weight of 46,274.01, while the *R. microplus* protein was 391 amino acids long with a molecular weight of 42,221.45. The *S. scabiei* protein had 36 positively charged residues (Arg + Lys) and 35 negatively charged residues (Asp + Glu), indicating its significance in potential interactions. On the other hand, the *R. microplus* protein exhibited 31 positively charged residues and 29 negatively charged residues, which still showcased its potential functional importance. The computed GRAVY (“Grand Average of Hydropathicity”) values further emphasized the distinct nature of these proteins. The *S. scabiei* protein displayed an incredibly low GRAVY score of 0.003, highlighting its highly hydrophilic nature. In contrast, the *R. microplus* protein showed a relatively higher GRAVY score of 0.105, signifying its better interaction with water and implying a more hydrophilic characteristic. To ascertain the stability of these proteins, the instability index (II) was computed. Remarkably, both target proteins exhibited stability with II values of 33.04 for *S. scabiei* and 32.30 for *R. microplus*, as shown in Table 7, both of which fall below the critical threshold of 50.

### 2.5. Secondary Structure Prediction

The structure of a protein is intimately linked to its function, and this connection is particularly evident in its secondary structure, which includes helices, sheets, turns, and coils. These elements play a crucial role in defining the protein’s overall structure, function, and interactions [29]. For the hypothetical protein, the secondary structure analysis, conducted using the SOPMA server, revealed the following percentages: 22.43% alpha helices, 40.57% random coils, 30.55% extended strands, and 6.44% beta-turns in the *S. scabiei* aspartic protease protein. On the other hand, the *R. microplus* aspartic protease protein exhibited 22.51% alpha helices, 40.15% random coils, 31.46% extended strands, and 5.88% beta-turns. Table 8 and the accompanying Figure 3 provide a clear representation of the representative secondary structures for both aspartic protease proteins in *R. microplus* and *S. scabiei*.

### 2.6. Functional Characterization

The SOSUI server effectively identified transmembrane regions for various proteins. By submitting a protein of interest, this server categorizes it as cytoplasmic or transmembrane. To enhance transmembrane helix prediction, an amphiphilic list of amino acid sequences was generated using the SOSUI server tool (https://harrier.nagahama-i-bio.ac.jp/sosui/, accessed on 22 May 2023). This framework, which combines amphiphilic amino acids, was utilized for isolating coat proteins and estimating transmembrane helical regions. For soluble proteins and membrane proteins, amino acid sequences were considered based on a sequence identity cutoff of 25%. Natural occurrences revealed the presence of lysine, arginine, tyrosine, and tryptophan amino acids near the ends of transmembrane helices. Amphiphilicity values are positive for polar residues with a long hydrophobic stem beyond the γ carbon, whereas amphiphilicity values are 0 for tiny polar and hydrophobic residues [30].

SOSUI server tool results show that both *S. scabiei* aspartic protease and *R. microplus* aspartic protease protein are in the form of membrane proteins. The transmembrane sections are rich in hydrophobic amino acids, with *S. scabiei* protein having a normal hydrophobicity of 0.003341 and *R. microplus* protein having a normal hydrophobicity of 0.104859. Other built-in SOSUI server applications, such as SOSUI (Batch), validated that both proteins are 100% membrane proteins. The SOSUI signal anticipated that both proteins would include a signal peptide. SOSUI gramN predicted that the cytoplasm is the subcellular localization site of *S. scabiei* aspartic protease while the subcellular localization site of *R. microplus* aspartic protease protein is outer membrane (Table 9, Figure 4).

### 2.7. Protein Model Building and Validation

When exclusively amino acid sequence data are provided, homology modeling may be used to predict protein structure. The protein structure is typically more important than the sequence alone in determining protein function. The biological principle behind homology modeling, also known as comparative modeling, is that a similarity in structure between two sequences predicts functional similarity. For sequence identities of more than 30% of a known structure, accurate low-resolution X-ray structure prediction is often achievable. In contrast, homology-based structure prediction can never be reliable with a sequence identity lower than 30% [31]. By using homology modeling on the SWISS-MODEL website, we were able to determine the three-dimensional structures of the aspartic protease proteins from *S. scabiei* and *R. microplus*. Only one model was generated with a GMQE of 0.88 and the best available template (2psg.1.A with 43.29% sequence identity) to build a 3-D model for *S. scabiei* aspartic protease and 0.68 GMQE (template; 5ux4.1.A with 54.99% sequence identity) for *Rhipicephalus microplus* aspartic protease. We utilized SAVESv6.0 to compare each model’s ERRAT value and Ramachandran plot, as shown in Figure 5. The Ramachandran plot analysis classified the residues into quadrilateral areas. Permitted regions are represented by yellow in the graph, whereas restricted regions are shown in red. For models created on the SWISS-MODEL server, PROCHECK produced a Ramachandran plot. After the models were refined, their stereochemical quality was determined by performing the Ramachandran map calculations with the help of the PROCHECK program. More than 85% of the residues in our examined proteins are found in the most widely dispersed region, demonstrating the precision and excellent quality of the modeled structure [32]. *S. scabiei* aspartic protease in the model has two residues in the forbidden area, but the same protein in *R. microplus* contains none. Validation with Ramachandran plot in SAVESv6.0 shows that the ERRAT value for the aspartic protease protein in *S. scabiei* is 82.703, whereas it is 80.719 for the aspartic protease protein in *R. microplus (*Figure 5A,B).

### 2.8. Active Sites Prediction

CASTp was used to find the key residues and region around the binding cavity of *S. scabiei* aspartic protease (LEU-47, ALA-50, LEU-52, GLY-53, SER-57, SER-58, ASP-61, SER-309, VAL-312, GLU-313, ASN-316, PRO-324, VAL-325, LYS-326, GLY-327, TYR-329, ILE-375and GLY-376) and *R. microplus* aspartic protease protein (ALA-69, GLN-70, TYR-71, VAL-87, ASP-89, SER-92, TYR-134, GLY-135, ALA-170, ALA-173, ALA-174, PHE-176, ILE-179, ASP-276, GLY-278, THR-279, VAL-356 and ILE-365) as shown in Figure 6.

### 2.9. Docking Analysis

The design of structure-based drugs is highly reliant on docking small molecule compounds into a receptor’s binding site and estimating the complex’s binding affinity. AutoDock Vina is a free and open-source molecular-docking, virtual-screening, and drug-discovery tool with multicore capability, lightning-fast processing rates, and an intuitive user interface. When the structure of the ligand–protein complex is known, the docking tool’s ability to imitate the ligand’s binding mode to the protein may be assessed. The root mean square deviation (RMSD) between the docked position and the ligand’s crystallographically observed binding site is widely used as a criterion, and a value less than 2 Å is typically regarded as successful.

CASTp was used to find the key residues and region around the binding cavity of the *S. scabiei* aspartic protease and the *R. microplus* aspartic protease protein. The active-site residues of both the protein making different numbers of hydrogen bonds and the hydrophobic bonds were identified. In this study, *S. scabiei* aspartic protease and *R. microplus* aspartic protease proteins were thus docked with the selected compounds to assess the binding interactions. Alpha-Amyrenyl acetate interacts with the *S. scabiei* receptor via a series of amino acid residues—the hydrogen bond (Tyr 329), pi-alkyl (Ala 50 and Tyr 329), and alkyl (Leu 47, Ile 375)—with the docking score of −7.3 Kcal/mol. The ranking of the docking score is as follows: Alpha-Amyrenyl acetate > Lupenol > Butelin > 3-Deoxyestradiol > cis-13,14-Epoxydocosanoic acid > Indole > alpha-Tocopherol > 5-Hydroxytryptamine > Ascorbyl 6-stearate > 3,4,4-Trimethyl-5-pyrazolone. Furthermore, the binding interactions of compounds and *R. microplus* aspartic protease proteins is as follows: Alpha-Tocopherol > Alpha-Amyrenyl acetate > 3-Deoxyestradiol > Lupenol > Butelin > 5-Hydroxytryptamine > Ascorbyl 6-stearate > Indole > cis-13, 14-Epoxydocosanoic acid > 3,4,4-Trimethyl-5-pyrazolone. Alpha-Tocopherol binds to the *R. microplus* aspartic protease protein via a series of bonds: hydrogen bond (Gly 139), pi-Alkyl and Alkyl (Tyr 134, Ile179, Ile379, Ala173, Ala 170, Ala 174, Phe 176, Val 87), Pi-Pi stacked(Tyr 134), and Pi-Anion (Asp276). The summaries of the docking study are shown in Figure 7.

### 2.10. Docking Validation

Our docking protocol underwent thorough validation across all target receptors, a crucial step taken to ensure the reliability of the molecular docking procedures and software. To validate the results, we employed Autodock Vina (version 1.1.2), and a potential grid box was created using AutoGrid4 with a spacing of 0.375 Å, approximately one-fourth of the length of a C-C covalent bond. The dimensions of the grid box were centered on X: 21.8344 Y: −5.6644 Z: 19.7712 Å for the *S. scabiei* aspartic proteinase protein and X: 21.6564 Y: −10.6987 Z: 22.5169 Å for the *R. microplus* aspartic proteinase protein. We carefully examined the interactions of the docking poses with the active site residues. Ultimately, we selected the pose with the highest binding affinity, which was determined to be −7.1 and −6.8 kcal/mol for *S. scabiei* aspartic protease and *R. microplus* aspartic protease, respectively. This validation exercise was deemed successful because the docked complexes precisely replicated the original ligand poses, matching the native ligands with RMSD values of 1.83 Å *S. scabiei* aspartic proteinase protein (Figure 8) and 1.04 Å for *R. microplus* aspartic proteinase protein (Figure 9).

## 3. Discussion

The necessity for tick management puts the dairy industry’s viability at risk in locations where ticks thrive and proliferate [34]. The same is the case for mite infestations. Chemical acaricides are often employed for this purpose, although [35] revealed that the development of acaricide resistance in tick species is a serious problem. Furthermore, the use of chemical acaricides may pollute the environment and contaminate cattle meat and milk, as well as promote tick and mite resistance [36]. In response to these issues, there is increased interest in the use of natural plant-based tick-management solutions. Many studies have looked at the efficiency of plant extracts and phytochemicals as acaricides, and the findings have been encouraging [25].

The current study revealed that the mean mortality of adult ticks was increased significantly with increased dosage (concentration) and exposure time after in vitro treatment for the tested botanicals. The same was the case for *S. scabiei* mites, whose mortality was also time- and dose-dependent. These results are in line with the findings of [37,38], in which the mortality effect of botanicals was indicated to be dose-, concentration- and exposure-time-dependent. Our study also revealed that all methanolic extracts of the tested botanical leaves at the tested concentrations induced a significant acaricidal effect against *S. scabiei* mites and *R. microplus* compared with the negative control.

Lead and target identification, followed by lead optimization, was a lengthy and costly process in the traditional drug development cycle. In the modern age, computational biology has created a low-cost, rapid method for finding novel drugs [39]. Potential therapeutic candidates for use against our target proteins are identified using molecular docking, an analysis of the likely binding affinities of two structures. The aspartic protease enzymes of *S. scabiei* mites and *R. microplus* were found to play an essential role in the survival and sustenance of the parasites by facilitating the breakdown of host hemoglobin during the blood-feeding process [27,40]. Therefore, preventing ticks and mites from producing aspartic protease may disrupt their blood-feeding process and perhaps restrict their survival and reproductive success by preventing them from digesting host hemoglobin. This method has the potential to be investigated as a means of decreasing the harm ticks and mites do to their hosts. Butelin, Lupenol, and Buterol are examples of phytochemicals. Alpha-Indole-3-deoxyestradiol, 13-cis-epoxydocosanoic acid, 14-epoxydocosanoic acid, tocopherol, 5HTP, Ascorbic Acid, and both 6-stearate and 3,4,4-trimethyl-5-pyrazolone isolated from *M. buxifolia* have been shown to meet all the requirements for classification as pharmaceuticals. Thus, the aforementioned phytochemicals were used in docking experiments with the aspartic protease parasite enzymes. Since we wanted to stop these specific enzymes from working, we bound inhibitors to their active sites. All of the phytochemicals were shown to have stronger affinities to the active pockets and lower binding energies compared to the synthetic drug. Furthermore, the existence of typical hydrogen bonds in the protein–ligand complex’s 2D bond contacts demonstrated robust binding. Aspartic protease inhibitory action was shown by plant-derived compounds with significant affinities, such as Alpha-Amyrenyl acetate against *R. microplus* and alpha-Tocopherol against *S. scabiei*.

## 4. Materials and Methods

### 4.1. Plant Collection, Identification and Extract Preparation

Aerial parts from *M. buxifolia* were collected from Abdul Wali Khan University Mardan’s (AWKUM) botanical garden in Khyber Pakhtunkhwa (coordinates: 34.1917° N, 72.0347° E). The plant materials were inspected for damage and rinsed. The leaves were then taken to the herbarium of the Department of Botany, AWKUM, and assigned the accession number Awkum.Bot. They were then air-dried for 2 weeks. After 15 days, the dried leaves were ground into powder, and stock solutions were prepared as described by [34]. The resulting solution was concentrated, yielding a stock extract for further dilutions and tests. For the mite bioassay, the stock solution was diluted to 0.25, 0.5, 1, and 2 g/mL, whereas for the adult immersion test (AIT) and the larval packet test (LPT), the concentrations were diluted to 2.5, 5, 10, 20, and 40 mg/mL concentrations.

### 4.2. Collection and Identification of Mites

Mites were collected from rabbits at AWKUM’s rabbit farms, where hay served as bedding and was changed daily. Upon detecting signs of mange, skin scraps were taken from the rabbits using the method described by [23] and immediate treatment was administered. The skin scraps were then examined under a compound microscope to identify *S. scabiei* var cuniculi.

### 4.3. Collection and Identification of Ticks

In strict adherence to the guidelines outlined by the World Association for the Advancement of Veterinary Parasitology [41] a careful manual collection of fully engorged *R. microplus* ticks was conducted from cattle and buffaloes at various farms located in Mardan, Khyber Pakhtunkhwa, with precise coordinates provided as 34.1986° N, 72.0404° E. Subsequently, the ticks underwent a comprehensive cleaning process and were morphologically identified as *R. microplus* through the utilization of standard tick identification keys, employing a microscope [42]. A total of 300 adult engorged female ticks were carefully selected for inclusion in the study. Some of these ticks were employed to obtain larvae for the larval packet test, while the remaining ticks were carefully divided into separate groups to conduct the adult immersion test, with a focus on evaluating the acaricidal effects of a fungal extract.

### 4.4. Contact Bioassay

The study was first approved by the ethical committee of AWKUM. Skin scraps were taken from the infested rabbits raised at the rabbit form adhering to the Guide for the Care and Use of Laboratory Animals (8th edition) [43]. The infested skin was first briefly cleaned and then scraped into a micro Petri plate using sterile surgical blade until the skin appeared red. The Petri plate containing skin scraps was incubated at 37 °C for the mites to emerge out of the skin scraps. The experimental setup involved the inoculation of ten mites into individual Petri plates using a fine needle. Following that, 0.5 mL samples of the plant extract were introduced directly onto the mites within the Petri plates. This procedure was carried out independently for each concentration of the extract, and each concentration was replicated three times.

### 4.5. Adult Immersion Test and Larval Packet Test

The extracts efficacy against *R. microplus* ticks was determined using the adult immersion test (AIT) and larval packet test (LPT) according to the protocol outlined by Ayub et al, [37] and Matos et al. [44]. The tick’s larval mortality was recorded at 24 and 48 h for different concentrations. Similarly, female ticks and their laid eggs were weighed in AIT. The effectiveness of the crude extracts in AIT was evaluated by calculating the percent inhibition of oviposition (% *IO*) using the following formula [45]:$$\%\;IO=\frac{\mathrm{Egg}\;\mathrm{laying}\;\mathrm{Index}\;\left(\mathrm{control}\right)-\;\mathrm{Egg}\;\mathrm{layingIndex}\;\left(\mathrm{treated}\right)}{\mathrm{Egg}\;\mathrm{laying}\;\mathrm{Index}\;\mathrm{control}}\times 100\%$$
where egg-laying index = mean weight of eggs laid ÷ mean weight of engorged females.

### 4.6. Sequence Retrieval

The amino acid sequences of aspartic protease protein from *S. scabiei* (accession numbers: V5NEJ5) and *R. microplus* aspartic proteinase protein with accession number C3UTE0 were obtained from the UniProtKB database. Both the protein sequences were retrieved in FASTA format and used for further analysis.

### 4.7. Characterization of the Physicochemical Properties

The protein’s physicochemical characterization required the calculation of many critical characteristics using well-established methodologies. Expasy’s ProtParam server [46] was used to calculate the theoretical isoelectric point (pI), molecular weight, total number of positive and negative residues, extinction coefficient [47], instability index [48], aliphatic index [49], and grand average hydropathy (GRAVY) [50]. The corresponding results can be found in the provided table.

### 4.8. Functional Characterization

In this study, the SOSUI server (https://harrier.nagahama-i-bio.ac.jp/sosui/, accessed on 22 May 2023 [51]) was employed to assess the amphiphilicity index and hydropathy index of the aspartic protease protein found in *R. microplus* and *S. scabiei* mite. These indices were used to estimate whether the protein was in the cytoplasm or bridged the transmembrane. Furthermore, the SOSUI server’s numerous tools, such as SOSUI (Batch), SOSUIsignal, SOSUIgramN, and SOSUImp1, were used to predict parts of the secondary structure of the aspartic protease protein from its supplied amino acid sequence [52]. This prediction helps in locating the hypothetical protein inside a cell.

### 4.9. Secondary Structure Prediction

The SOPMA tool (https://npsa-prabi.ibcp.fr/cgi-bin/npsa_automat.pl?page=/NPSA/npsa_sopma.html, accessed on 22 May 2023 [27]) was utilized to compute the secondary structural characteristics of the protein sequences chosen for this investigation. The obtained results are comprehensively reported in Table 8.

### 4.10. Building and Evaluation of the 3D Structure Model 

The FASTA sequences of the target proteins were submitted to the SWISS-MODEL server (https://swissmodel.expasy.org/, accessed on 25 May 2023) for prediction of the 3D structure model using automated comparative modeling as described by [53]. To ensure the reliability of the predicted 3D structure model for the hypothetical protein, multiple quality assessment tools were employed. Firstly, the 3D model quality was assessed using SAVES (https://saves.mbi.ucla.edu/, accessed on 26 May 2023, which is a comprehensive platform for protein structure validation. The Ramachandran plot was constructed using the PROCHECK server [54] to visualize the distribution of backbone dihedral angles (ψ against φ) for all amino acid residues in the protein structure. The analysis of this plot helps to evaluate the stereochemical quality and identify any potential deviations from ideal conformations in the model. Furthermore, the protein structure was cross-validated using the ERRAT server (https://services.mbi.ucla.edu/ERRAT, accessed on 1 June 2023). This tool assesses the statistics of nonbonded interactions between different atom types within a 9-residue sliding window [55]. The plot of the error function versus the window position provides valuable insights into the accuracy and consistency of the protein crystallographic structure. By integrating the results from these various assessment methods, the reliability and accuracy of the predicted 3D structure model for the hypothetical protein were thoroughly verified.

### 4.11. Active Site Determination

The use of the Computed Atlas of Surface Topography of Proteins (CASTp) server (http://sts.bioe.uic.edu/castp/, accessed on 2 June 2023) has proven to be an immensely valuable and indispensable tool for determining the active sites of proteins. Through CASTp, a highly detailed, comprehensive, and quantitative analysis of the topographical features of proteins is achieved. This analysis enables the precise localization and measurement of active pockets on both the protein’s surface and its interior 3D structure. As a result, CASTp facilitates the accurate prediction of critical regions and key residues within the protein that play a significant role in interacting with ligands [56]. To enhance the understanding and interpretation of the CASTp results, the visualization of the outcomes has been effectively accomplished using the PyMOL (version 2.5.4) software.

### 4.12. Ligand Preparation

The thirteen structures of chemical constituents of *M. buxifolia*, Alpha-Amyrenyl acetate, Lupenol, Butelin, 3-Deoxyestradiol, cis-13,14-Epoxydocosanoic acid, Indole, Alpha-Tocopherol, 5-Hydroxytryptamine, Ascorbyl 6-stearate and 3,4,4-Trimethyl-5-pyrazolone, were collected from published literatures [21,57]. The ligands’ 2D chemical structures were drawn using ChemDraw Ultra 2008, and their energy minimization was performed using Chem3D Ultra. The resulting ligand structures were saved in .pdb format.

### 4.13. Molecular Docking Analysis

The binding mode and interaction of target proteins with individual chemical constituents of *M. buxifolia* were thoroughly investigated using AutoDock Vina software (version 1.1.2). The docking process aimed to explore a range of possible conformations and orientations for the ligands at the binding sites. Prior to docking, the protein structures were prepared in PyRx (version 0.8) software, generating PDBQT files with added hydrogen atoms to all polar residues. The ligands’ bonds were set to be rotatable to enhance flexibility during docking. To carry out the protein-fixed ligand-flexible docking, the Lamarckian Genetic Algorithm (LGA) method was employed for all calculations. The binding site on the target proteins, *S. scabiei* aspartic proteinase protein and *R. microplus* aspartic proteinase protein, was defined by establishing a grid box with a grid spacing of 0.375 Å. For the *S. scabiei* aspartic proteinase protein, the grid box was centered on coordinates X: 21.8344, Y: −5.6644 and Z: 19.7712 Å, while for the *R. microplus* aspartic proteinase protein, it was centered on coordinates X: 21.6564, Y: −10.6987 and Z: 22.5169 Å. To ensure the reliability of the results, ten runs with AutoDock Vina were performed for each ligand structure in all cases, saving the best pose from each run. The final affinity value was determined by averaging the affinity of the best poses obtained in the ten runs. The analysis of protein–ligand conformations, including interactions such as hydrogen bonds and bond lengths, was conducted using PyMol (version 2.5.4) and Discovery Studio Visualizer (version 4.5), allowing for a comprehensive examination of the complex structures and their interactions.

### 4.14. In Silico Docking Protocol Validation

The validation of our docking protocol aimed to establish its precision and reliability in replicating the binding model and molecular interactions observed in experimentally modeled protein structures during our current in silico studies. To validate the procedure, we utilized an aspartic protease inhibitor as a test case. The validation process involved manually removing modeled proteins and redocking the inhibitor into the active site using AutoDock Vina (version 1.1.2) software. This entailed extracting inhibitor heteroatoms from the protein complex, saving them as a separate inhibitor in PDB format, while keeping the grid parameters consistent. This rigorous validation ensured the accurate binding of the inhibitor to the active site cleft, with minimal deviation compared to the actual co-crystallized complex. Subsequently, we superimposed the redocked complex onto the reference co-crystallized complex using PyMOL 2.3, calculated the root mean square deviation (RMSD), and generated a 2-dimensional image highlighting the relevant amino acid residues with Discovery Studio (version 4.5) software. This process ensured the reliability of our docking methodology and its fidelity in reproducing binding characteristics observed in experimental crystallographic data.

### 4.15. Statistical Analysis

All statistical approaches were made in R and RStudio. The data were first arranged in Microsoft Excel (v 2302) and then imported into the R working environment for further statistical analysis. Descriptive statistics of the data were calculated and presented in a table with mean ± standard deviation. The significant difference between the different concentrations were calculated using one-way ANOVA (analysis of variance) followed by the Tukey honesty significance difference (HSD) test. Furthermore, 50% and 90% lethal concentration and lethal time (LC_50_, LC_90_ and LT_50_, LT_90_) were calculated in R using the “ecotox” package, and all the data were graphically presented using “ggplot2 and ggpubr” R package.

## 5. Conclusions

Based on the study conducted, we can confidently assert that the *M. buxifolia* plant possesses notable acaricidal properties, effectively eliminating ticks and mites. Moreover, our computational approach has identified promising phytochemicals with the potential to serve as viable drug candidates against *S. scabiei* and *R. microplus*. These findings warrant further investigation through clinical trials, paving the way for the development of an effective acaricidal drug. The robust efficacy of the most potent chemicals within the plant extract positions them as integral components of a comprehensive strategy to combat *R. microplus* ticks and *S. scabiei* mites.

## Figures and Tables

**Figure 1 molecules-28-06930-f001:**
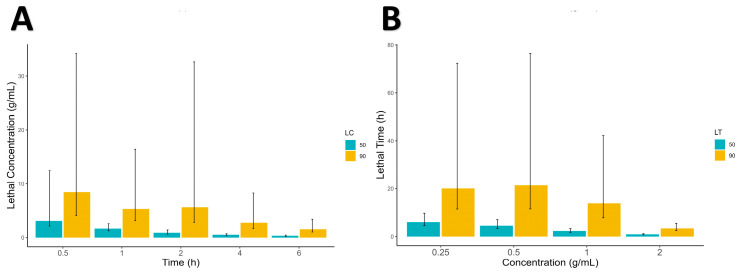
The lethal concentration (**A**) and lethal time (**B**), along with its confidence limits as the error bar, for *M. buxifolia* extract against *S. scabiei* in vitro.

**Figure 2 molecules-28-06930-f002:**
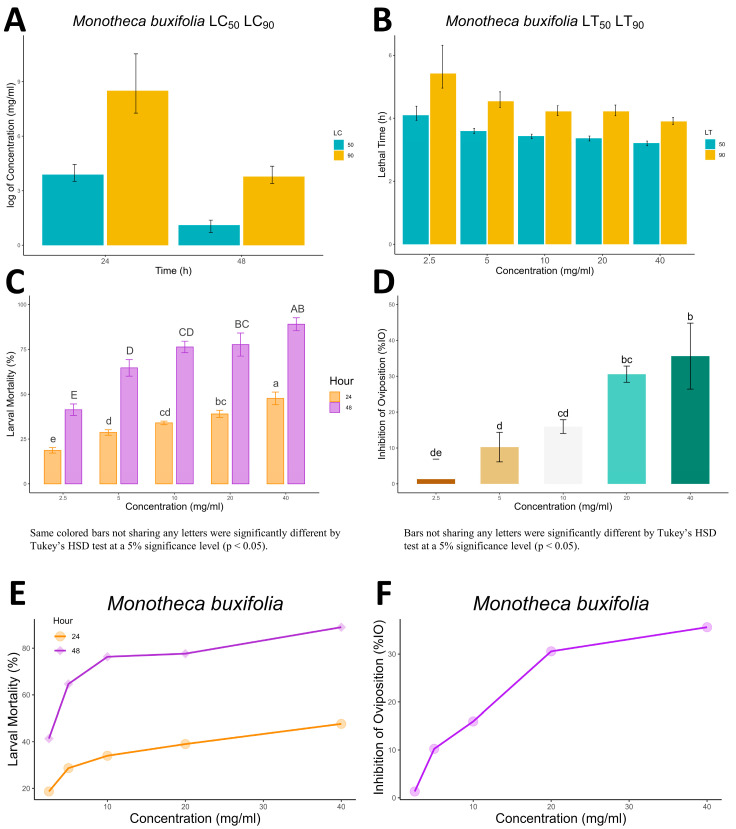
(**A**,**B**) show the LC and LT, respectively, of *M. buxifolia* extract against *R. microplus* along with its lower and upper confidence limits as error bars. (**C**,**D**) represent the significant difference between the extract’s concentration for larval mortality and %IO, respectively whereas, the capital letters in (**C**) represent the significant differences between the larval mortality values at 48 h for different concentrations while the small letters represent the same for 24 h. (**E**,**F**) show the concentration vs. response graph for larval mortality and %IO, respectively.

**Figure 3 molecules-28-06930-f003:**
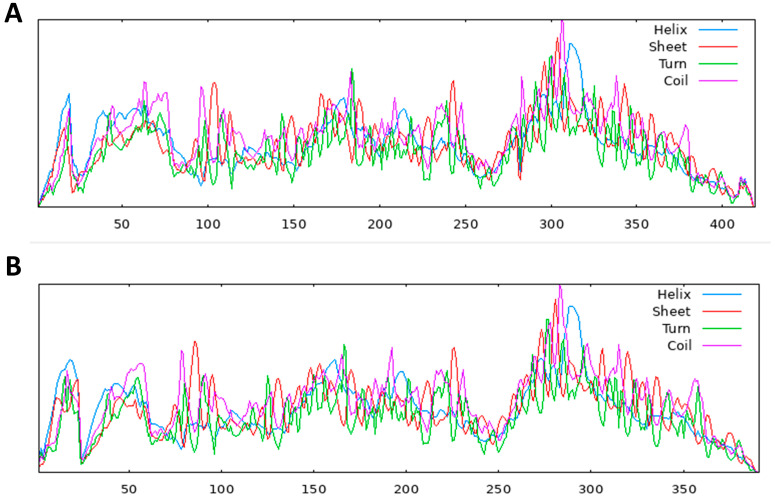
Secondary structural models, (**A**) *S. scabiei aspartic protease* protein and (**B)**
*R. microplus aspartic protease* secondary structure, predicted from multiple alignments using the SOPMA server.

**Figure 4 molecules-28-06930-f004:**
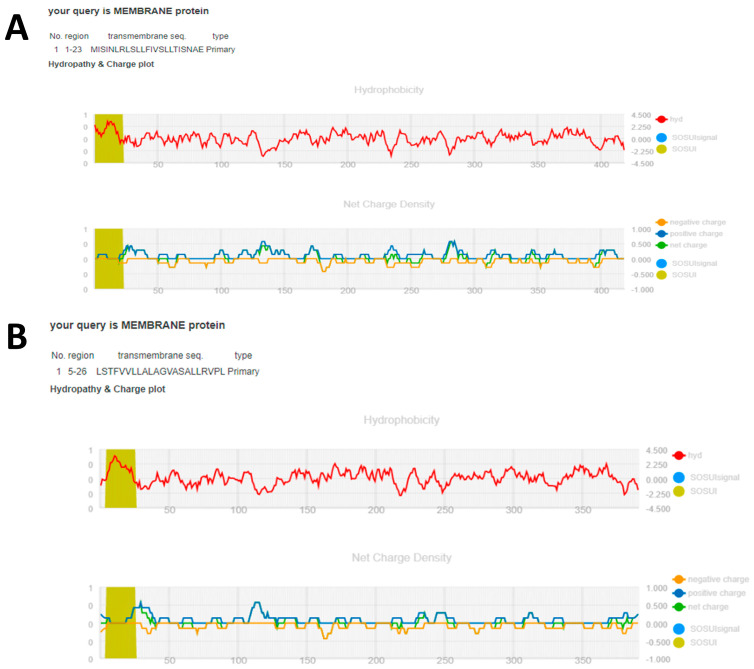
Hydropathy and charge plots of (**A**) *S. scabiei* aspartic protease and (**B**) *R. microplus* aspartic protease protein from the SOUSI server.

**Figure 5 molecules-28-06930-f005:**
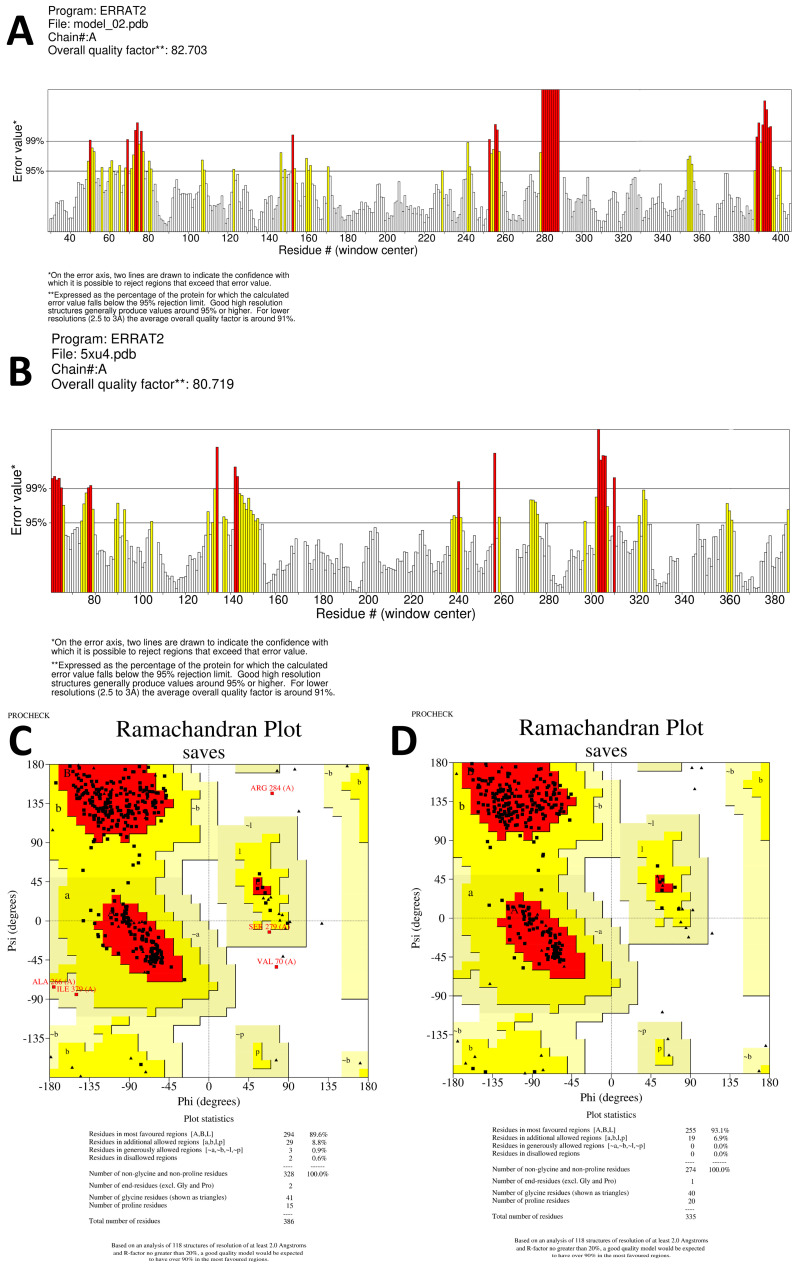
(**A**,**C**) Shows the SAVES server’s ERRAT plot and Ramachandran plot, respectively for the validation of *S. scabiei* aspartic protease whereas, (**B**,**D**) represents the validation of *R. microplus* aspartic protease through ERRAT servers plot and Ramachandran plot, respectively.

**Figure 6 molecules-28-06930-f006:**
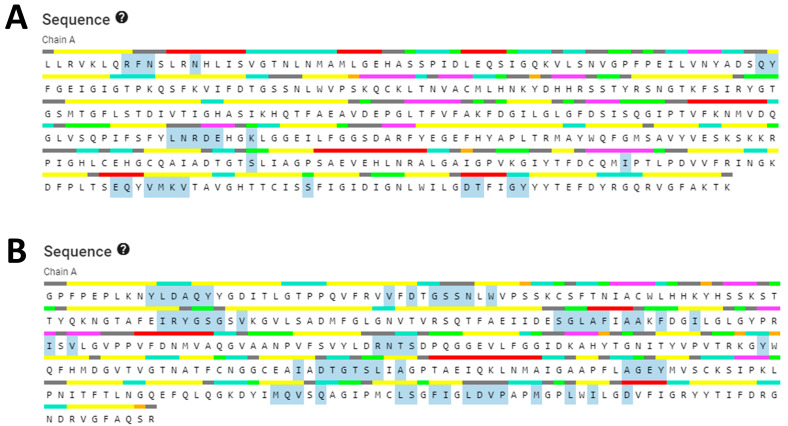
The active sites predicted using the CASTp server for (**A**) *S. scabiei* aspartic protease (*SsAP*) and (**B**) *R. microplus* aspartic protease (*RmAP*).

**Figure 7 molecules-28-06930-f007:**
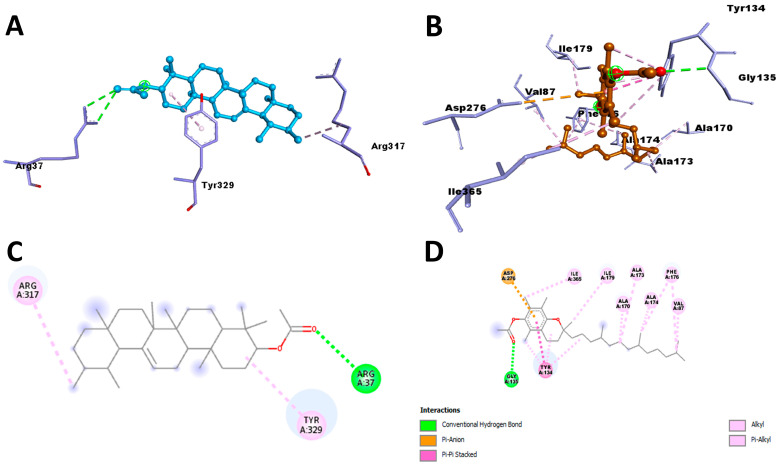
(**A**,**C**) show the 3D and 2D interaction of Alpha-Amyrenyl acetate with *Sarcoptes scabiei* receptor, respectively, whereas (**B**,**D**) show the 3D and 2D interaction of Alpha-Tocopherol with *R. microplus* aspartic protease protein, respectively. The sky blue color (**A**) represents the ligand wheras the other color represents the protein in.

**Figure 8 molecules-28-06930-f008:**
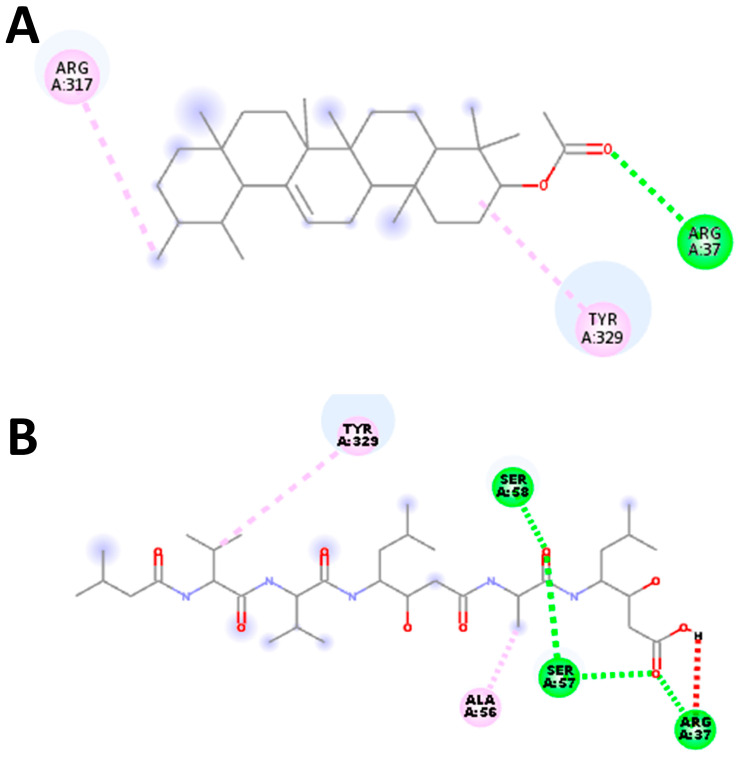
Protocol validation of molecular docking experiment with *S. scabiei* aspartic protease using AutoDock Vina and Discovery studio. Comparison of binding modes for re-docked ligand (**A**) vs. reference compound pepstatin (5FP) [33] (**B**). Amino acid residues’ interaction with (**B**) standard drug and (**A**) re-docked ligand, accomplished in Discovery studio.

**Figure 9 molecules-28-06930-f009:**
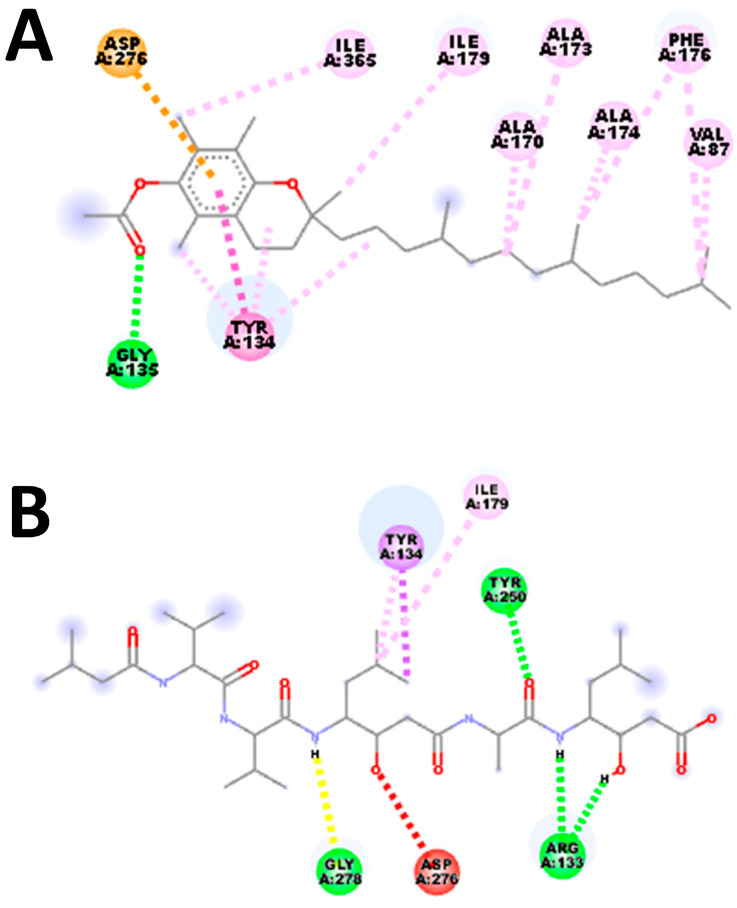
Protocol validation of the molecular docking experiment with *R. microplus* aspartic protease protein using AutoDock Vina and Discovery studio. (**A**) Comparison of binding modes for re-docked ligand (**A**) vs. reference compound pepstatin (5FP) [31] (**B**). Amino acid residues’ interaction with (**B**) standard drug and (**A**) re-docked ligand, accomplished in Discovery studio.

**Table 1 molecules-28-06930-t001:** The average % mortality ± standard deviation (SD) of *S. scabiei* at different concentrations of *M. buxifolia* plant extract.

Plant	Concentration (g/mL)	*n*	Time Interval (h)				
			0.5	1	2	4	6
*Monotheca buxifolia*	2	3	26.67 ± 05.77 ^a^	56.67 ± 05.77 ^a^	73.33 ± 05.77 ^a^	90 ± 0 ^a^	100 ± 0 ^a^
	1	3	10 ± 0 ^b^	30 ± 0 ^b^	50 ± 0 ^b^	63.33 ± 05.77 ^ab^	73.33 ± 05.77 ^a^
	0.5	3	0 ± 0 ^c^	13.33 ± 05.77 ^d^	33.33 ± 11.55 ^d^	43.33 ± 11.55 ^c^	56.67 ± 05.77 ^c^
	0.25	3	0 ± 0 ^c^	0 ± 0 ^c^	20 ± 0c ^d^	33.33 ± 05.77 ^c^	46.67 ± 05.77 ^c^
Control Group	Permethrin	3	06.67 ± 05.77 ^bc^	26.67 ± 05.77 ^b^	43.33 ± 05.77 ^bc^	73.33 ± 05.77 ^b^	90 ± 0 ^b^
	Distilled Water	3	0 ± 0 ^c^	0 ± 0 ^d^	0 ± 0 ^e^	0 ± 0 ^d^	10 ± 0 ^d^

Means with different letters in the same column are significantly different according to Tukey’s HSD test at a 5% level of significance (*p*  <  0.05) between the experimental plant extract and control.

**Table 2 molecules-28-06930-t002:** Medium lethal concentration causing 50% and 90% mortalities (LC50 and LC90 values) of *M. buxifolia* plant extract at varying time intervals against *S. scabiei* in vitro.

Time (h)	LC_50_ (g/mL)	95% Confidence Limits		LC_90_ (g/mL)	95% Confidence Limits		Slope ± SE	Intercept ± SE	Chi Square (χ2)	*p*-Value
		LCL	UCL		LCL	UCL				
0.5	3.116	2.129	12.435	8.453	4.113	34.213	2.957 ± 0.946	−1.46 ± 0.241	1.024	1
1	1.642	1.256	2.559	5.325	3.179	16.4	2.508 ± 0.505	−0.54 ± 0.142	2.13	0.995
2	0.898	0.635	1.405	5.641	2.814	32.677	1.606 ± 0.373	0.075 ± 0.13	1.733	0.998
4	0.526	0.35	0.719	2.737	1.663	8.291	1.789 ± 0.383	0.499 ± 0.141	3.47	0.968
6	0.342	0.202	0.467	1.529	1.037	3.429	1.971 ± 0.421	0.918 ± 0.166	5.804	0.831

**Table 3 molecules-28-06930-t003:** Medium lethal time causing 50% and 90% mortalities (LT50 and LT90 values) of *M. buxifolia* plant extract at different concentrations against *S. scabiei* in vitro.

Concentration (g/mL)	LT_50_ (h)	95% Confidence Limits		LT_90_ (h)	95% Confidence Limits		Slope ± SE	Intercept ± SE	Chi Square	*p*-Value
		LCL	UCL		LCL	UCL				
2	0.931	0.673	1.192	3.356	2.474	5.479	2.301 ± 0.358	0.072 ± 0.133	2.814	0.999
1	2.338	1.719	3.301	13.878	7.894	42.191	1.657 ± 0.299	−0.611 ± 0.145	1.404	1
0.5	4.471	3.316	7.134	21.391	11.545	76.451	1.885 ± 0.353	−1.226 ± 0.189	5.913	0.949
0.25	5.967	4.522	9.821	20.05	11.514	69.739	2.435 ± 0.501	−1.889 ± 0.298	3.364	0.996

**Table 4 molecules-28-06930-t004:** Mean % larval mortality and %IO at different concentrations of *M. buxifolia* plant extract against *R. microplus*.

Plant	Concentration (mg/mL)	n	Mean % Mortality ± S.D		Mean %IO ± S.D
			24 h	48 h	
*Monotheca buxifolia*	40	3	47.667 ± 3.512 ^a^	89 ± 3.606 ^ab^	35.612 ± 9.210 ^b^
	20	3	39 ± 2 ^bc^	77.667 ± 6.429 ^bc^	30.574 ± 2.261 ^bc^
	10	3	34 ± 1 ^cd^	76.333 ± 3.215 ^cd^	15.960 ± 1.920 ^cd^
	5	3	28.667 ± 1.528 ^d^	64.667 ± 4.619 ^d^	10.239 ± 4.122 ^d^
	2.5	3	18.667 ± 1.528 ^e^	41.333 ± 3.215 ^e^	1.326 ± 5.569 ^de^
Control	Permethrin 5% (*w*/*w*)	3	42.667 ± 1.528 ^ab^	91.333 ± 4.509 ^a^	54.755 ± 8.611 ^a^
	Distilled Water	3	1.333 ± 1.528 ^f^	4.667 ± 3.786 ^f^	−6.386 ± 2.160 ^e^

Means not sharing any letters in the same column were significantly different according to Tukey’s HSD test at a 5% significance level (*p* < 0.05); S.D: standard deviation, n: number of replicates.

**Table 5 molecules-28-06930-t005:** Medium lethal concentration causing 50% and 90% mortalities (LC50 and LC90 values) of *M. buxifolia* plant leaf extract against *R. microplus* in vitro.

Time (h)	LC_50_ (mg/mL)	95% Confidence Limits		LC_90_ (mg/mL)	95% Confidence Limits		Slope ± S.E.	Intercept ± S.E.	Chi Square	*p* Value
		LCL	UCL		LCL	UCL				
24	48.678	33.689	85.498	4925.155	1437.592	37,437.56	0.639 ± 0.081	-1.078 ± 0.09	3.689	0.994
48	3.013	2.022	3.988	43.759	30.086	77.304	1.103 ± 0.087	-0.528 ± 0.087	24.78	0.025

LCL: low confident limit, UCL: upper confident limit, S.E.: standard error.

**Table 6 molecules-28-06930-t006:** Lethal time causing 50% and 90% mortalities (LT50 and LT90 values) at varying concentrations for *M. buxifolia* against *R. microplus* in vitro.

Concentration (mg/mL)	LT_50_ (h)	95% Confidence Limits		LT_90_ (h)	95% Confidence Limits		Slope ± S.E.	Intercept ± S.E.	Chi Square	*p* Value
		LCL	UCL		LCL	UCL				
2.5	60.179	51.172	80.082	226.023	142.238	554.957	2.23 ± 0.369	−3.968 ± 0.575	1.16	0.885
5	36.363	33.659	39.521	93.603	76.967	126.4	3.121 ± 0.355	−4.871 ± 0.547	2.096	0.718
10	30.913	28.8	33.004	67.871	59.667	81.415	3.752 ± 0.362	−5.591 ± 0.554	1.233	0.873
20	28.909	26.604	31.061	67.901	59.056	83.215	3.456 ± 0.362	−5.049 ± 0.552	5.102	0.277
40	24.77	22.808	26.492	49.446	45.073	56.056	4.269 ± 0.4	−5.95 ± 0.598	3.645	0.456

LCL: low confident limit, UCL: upper confident limit, S.E.: standard error.

**Table 7 molecules-28-06930-t007:** Physicochemical properties of the protein through Expasy ProtParam server.

Protein Name	Number of Amino Acids	Molecular Weight	Theoretical pI	Negatively Charged Residues (Asp + Glu)	Positively Charged Residues (Arg + Lys)	Total Number of Atoms	Instability Index	Aliphatic Index	Grand Average of Hydropathicity
**SsAP**	419	46,274.01	7.68	35	36	6501	33.04	89.33	0.003
**RmAP**	391	42,221.45	8.13	29	31	5927	32.30	86.52	0.105

SsAP: *Sarcoptes scabiei* aspartic protease, RmAP: *R. microplus* aspartic protease.

**Table 8 molecules-28-06930-t008:** Data of secondary structures predicted from multiple alignments by SOPMA server.

Protein Name	Alpha Helix	3_10_ Helix	Pi Helix	Beta Bridge	Extended Strand	Beta Turn	Bend Region	Random Coil	Ambiguous States	Other States
**SsAP**	22.43%	0.00%	0.00%	0.00%	30.55%	6.44%	0.00%	40.57%	0.00%	0.00%
**RmAP**	22.51%	0.00%	0.00%	0.00%	31.46%	5.88%	0.00%	40.15%	0.00%	0.00%

SsAP: *Sarcoptes scabiei* aspartic protease, RmAP: *R. microplus* aspartic protease.

**Table 9 molecules-28-06930-t009:** SOSUI results of *S. scabiei* aspartic protease and *R. microplus* aspartic protease protein.

Proteins in SOSUI Server	Region	Transmembrane Seq	Type	SOSUI (Signal)	SOSUI gramN (Subcellular Localization Site)	SOSUI (Batch)
**SsAP**	1–23	MISINLRLSLLFIVSLLTISNAE	Primary	signal peptide	Cytoplasmic	membrane protein
**RmAP**	5–26	LSTFVVLLALAGVASALLRVPL	Primary	signal peptide	Outer membrane	membrane proteins

SsAP: *S. scabiei* aspartic protease, RmAP: *R. microplus* aspartic protease.

## Data Availability

Not applicable.

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
