# Peer review of "Integrating In Vitro and In Silico Approaches to Assess Monotheca buxifolia Plant Extract against Rhipicephalus (Boophilus) microplus and Sarcoptes scabiei"

_molecules, 2023, doi:10.3390/molecules28196930_

Round 1
Reviewer 1 Report
The authors of the paper found that the M. buxifolia plant extract can be used against Rhipicephalus (Boophilus) microplus and Sarcoptes scabiei. The manuscript is overall well written and has good organization with minor English language and style spell check required. The authors have done a great job on analyzing the experimental data. Reconsider after major revision
Major points:
1. The authors should include more recent updates on the topic and compare how this study advances current knowledge in the "Introduction" section.
2. The "Conclusion" section should be described in more detail.
One small question.
“The positive control permethrin and negative control distilled water” What is meant by a positive control or a negative control?
Author Response
We have revised the manuscript accordingly. Please find attached word file.

Reviewer 2 Report
The manuscript "Integrating in vitro and in silico approaches to assess Monotheca buxifolia plant extract against Rhipicephalus (Boophilus) microplus and Sarcoptes scabiei" has serious flaws and must be rejected.
The results showed the values expressed in unusual g/mL (in Table. 1, Table 3, for example).
The concentration of permethrin was not indicated in Tables
The error bars were absent in Figure 1A and are extremely elevated in Figure 1C and Figure 1D. The variation is better than the observed values, showing the low quality of experimental conditions in which the assays were made.
The target (Aspartic protease) proposed is speculative and must be removed or justified better. In silico analysis must use an Aspartic protease inhibitor in redocking the template used to get the model and used as a possible inhibitor of the proposed target.
A biochemistry validation that the compounds target Aspartic protease must be made, followed by enzyme kinetics to prove that compounds are a competitive inhibitor of the target enzyme. If the compound shows, for example, a noncompetitive mechanism of inhibition, the docking does not make sense.
Author Response
We have revised the manuscript according to the reviewer's comments. Please find attached the response sheet. Thanks

Reviewer 3 Report
Dear authors
The research and the results were interesting, however should be organized and not repetitives (selection the best form for show withouth repeat the information). See more comments inside of the document

Author Response
We have revised the manuscript accordingly. Please find attached the response sheet. Thanks

Round 2
Reviewer 1 Report
I recommend accept in present form.
Author Response
Thank you.
Reviewer 2 Report
The article has serious flaws, additional experiments are needed, and research is not conducted correctly.
For example, Table 1 shows several data with error zero. On the other hand, figure I show an error bar greater than the values. The most of results shown in the table use millesimal when the errors are in the unit. The scientific accuracy of showing results must be considered.
The first LC50 value (TABLE 2) is reported to be 3 g/mL (3000 mg/mL). The extract was soluble at this concentration?
Also in Table 2, the authors showed slope values and intercepts: Did they calculate the LC50 and LC90 with linear regression?
Author Response
Thanks for the reviewer's kind comments and suggestions. We have tried and answered the comments. Please find attached a response sheet of the comments

Reviewer 3 Report
The document has substantially improved
Author Response
Thank You